# The Role of the Gallbladder, the Intestinal Barrier and the Gut Microbiota in the Development of Food Allergies and Other Disorders

**DOI:** 10.3390/ijms232214333

**Published:** 2022-11-18

**Authors:** Ana G. Abril, Tomás G. Villa, Ángeles Sánchez-Pérez, Vicente Notario, Mónica Carrera

**Affiliations:** 1Department of Microbiology and Parasitology, Faculty of Pharmacy, University of Santiago de Compostela, 15706 Santiago de Compostela, Spain; 2Sydney School of Veterinary Science, Faculty of Science, University of Sydney, Sydney, NSW 2006, Australia; 3Department of Radiation Medicine, Lombardi Comprehensive Cancer Center, Georgetown University, Washington, DC 20057, USA; 4Department of Food Technology, Spanish National Research Council, Marine Research Institute, 36208 Vigo, Spain

**Keywords:** gastrointestinal tract, gallbladder, microbiota, lupus erythematosus, leaky gut, polyamines, food allergies

## Abstract

The microbiota present in the gastrointestinal tract is involved in the development or prevention of food allergies and autoimmune disorders; these bacteria can enter the gallbladder and, depending on the species involved, can either be benign or cause significant diseases. Occlusion of the gallbladder, usually due to the presence of calculi blocking the bile duct, facilitates microbial infection and inflammation, which can be serious enough to require life-saving surgery. In addition, the biliary salts are secreted into the intestine and can affect the gut microbiota. The interaction between the gut microbiota, pathogenic organisms, and the human immune system can create intestinal dysbiosis, generating a variety of syndromes including the development of food allergies and autoimmune disorders. The intestinal microbiota can aggravate certain food allergies, which become severe when the integrity of the intestinal barrier is affected, allowing bacteria, or their metabolites, to cross the intestinal barrier and invade the bloodstream, affecting distal body organs. This article deals with health conditions and severe diseases that are either influenced by the gut flora or caused by gallbladder obstruction and inflammation, as well as putative treatments for those illnesses.

## 1. Introduction

The gut microbiota is currently considered a “hidden metabolic organ”, capable of controlling distal human tissues via the release of chemical mediators [1,2]. Even the earliest known humans understood that while some foodstuffs produced beneficial effects on their bodies, others could be detrimental to their health. Long before the discovery of microorganisms, microbes were already used in the treatment of human diseases. A prime example is the use of a preparation containing fecal matter from healthy humans (known as “yellow soup”), in ancient China, to successfully treat dangerous watery diarrheas; this, in turn, probably represents the first documented practice of the therapy currently known as “microbial fecal transplantation” [3]. The consumption of fermented dairy products containing bacterial species belonging to the genera *Lactobacillus*, *Lactococcus* or *Brevibacterium* is another early strategy used by a wide variety of human cultures to exploit the beneficial effects of microorganisms on human health; these microbes are known to prevent diseases such as urogenital bacterial infections [4]. In addition, it has long been known that wide variations in the gut microbiota (dysbiosis) are associated with certain diseases [1,5]; thus, it is no wonder that the intestinal microbiota is a major target for modern therapeutics. Food allergies have considerably increased since the end of the 20th century, and their ongoing prevalence is a cause for concern. For instance, food allergies are currently estimated to affect 8% of children and 11% of adults in the United States, while allergy to peanuts, which is believed to have affected 0.4% of people in 1997, already impacted 1.4% of the population by 2008 [6,7].

Food-related allergic conditions can be IgE-mediated (type I hypersensitivity), with their manifestations occurring within minutes of ingestion, with symptoms including hives and redness of the skin, vomiting and diarrhea, and even progressing to anaphylaxis in severe cases. On the other hand, some food allergies appear not to be mediated by IgE and involve time-delayed symptoms that can develop within days, or even hours. In fact, many researchers consider “non-IgE-mediated” allergic reactions as an umbrella term that includes those allergic pathologies for which there is no direct association between food consumption and the development of allergy. Allergic reactions to either milk or soy proteins are typical examples of this type of delayed food allergy, which, at least in children, rarely results in life-threatening responses such as the perilous anaphylactic shock. Normal symptoms include abdominal pain, vomiting and diarrhea with colic attacks, although it can also involve constipation. As is to be expected, the diagnosis of non-IgE-mediated food allergies is difficult and time-consuming, as it requires the withdrawal of particular food items from the diet, as removal of the food that triggers the allergic reaction should eliminate the pathological signs, with the symptoms re-appearing when the harmful food item is reintroduced into the diet. Both IgE-mediated and non-IgE-mediated food allergies can coexist in patients, although they entail different treatments; while a serious case of the first type of food allergy (anaphylaxis) requires adrenaline therapy, this hormone is not used in the treatment of allergic reactions not involving IgE. In addition, non-IgE-mediated food allergies usually affect particular areas of the gastrointestinal (GI) tract; these include: (i) the jejunum, or the middle part of the small intestine, as is the case for celiac disease, which is a long-term autoimmune disorder (the clinical symptoms of this illness had already been described by 250 A.D. [8]; (ii) the esophagus (gullet), causing eosinophilic esophagitis, also known as EoE, resulting in a leaky esophageal epithelium and inducing the T-helper 2 pathway [9]; (iii) the small and large intestines, producing the food-protein-induced enterocolitis syndrome (FPIES) [10,11,12]; and (iv) the large intestine, resulting in proctocolitis, which is an inflammation of the rectum and colon [13,14].

The accumulation of unfavorable microorganisms in certain parts of the human body, such as the gallbladder, can either contribute to the development of new food allergies, or exacerbate existing ones. On the other hand, certain types of beneficial bacteria, either representing existing gut flora or supplied as probiotics, can ameliorate a variety of food allergies, as it is well known that there is no cure for food allergies. Recent research indicates that a breach in the intestinal epithelium integrity, either by the action of harmful gut bacteria or as a consequence of genetic disorders, generates the release of cell mediators (including IgE), resulting in food allergies.

## 2. The Human Gut and Gallbladder

The human gut, as is the case for the gastrointestinal tract in animals, represents a “microbiologically open system” with apertures at both ends, that is composed of extremely high numbers of bacteria, archaea, fungi, viruses (including bacteriophages), and protozoa. This open system not only represents an extraordinary example of microbial mutualism and parasitism, but it must also perform complex biochemical procedures in order to accomplish food digestion. It is generally accepted that, over the course of a human lifetime, approximately 60 tons of food pass through the GI tract, which has a vast surface area, appraised as between 200 and 400 m^2^, that is dedicated to nutrient adsorption [15]. Although a few years ago it was estimated that the gut microorganisms outnumbered the cells in the human body by a factor of 10:1, more recent calculations suggest that the ratio for GI bacteria is closer to 1:1 [16]. The gut microbiota affects many aspects of human health, but the present review will mainly concentrate on the microbial effects on the human immune system [17,18]. Occasionally, dysbiosis occurs, resulting in a variety of syndromes, which can include the development of food allergies. From birth, microbes colonize a variety of econiches in the human body, including the skin, hair, mouth, vagina, and the GI tract [19]. As mentioned above, it is generally accepted that the gut microbiota represents an average of 1.5–2.0 kg of fecal matter [20], and it is estimated to include 3 million microbial genes, which constitute the equivalent of 150 times the number of human genes [21,22]. Using the Illumina-based metagenomic sequencing they developed, Qin and co-workers reported in 2010 the characterization of 3.3 million non-redundant microbial genes (from fecal samples provided by 127 individuals of European ancestry), representing mainly bacterial genes (*ca*. 99%); the authors concluded that the human gut is overwhelmingly dominated by bacterial genomes that are ~150 times larger than the human genome.

According to Qin and co-workers [21], the GI microbiota is not homogeneous among human beings, with an estimated high percentage (up to 60%) of microbes specific to particular individuals. However, the microbiota undergoes alterations at different stages in life, as it is influenced by factors such as age, environment, drug usage, antibiotic treatment, and diet. Despite individual variations, the Gram-positive bacteria present in human GI flora mainly correspond to species within the Firmicutes and Actinobacteria phyla, while most of the Gram-negative bacteria belong to the groups Proteobacteria and Bacteroidetes. In fact, in healthy people, Firmicutes and Bacteroidetes constitute the predominant bacterial phyla present in the gut [1]. Unfortunately, the components of the non-bacterial intestinal microbiota, including fungi, protozoa, viruses, and viroids, are, in general, not so well known.

A main characteristic of the lymphatic system is that it contains microorganisms (bacteria), that can use the lymph vessels to travel to different areas of the body [23]. In addition to the lymph, bacteria can also colonize the secondary lymphoid organs, which include the lymph nodes, spleen, intestinal Peyer’s patches, nasal adenoids, and tonsils. Additional human organs, such as the gallbladder, can also harbor microorganisms, which probably originate from the bacteria in the saliva as well as from microbes in the upper GI tract, although no definitive results are available.

The gallbladder has long been recognized as a crucial member of the human digestive system. It is a small sac-shaped greenish organ attached to and located just beneath the liver, and its role is to store hepatic bile. Not all mammals have a gallbladder (it is absent in species such as rats, pigeons and horses) but, as a general rule, it is present in omnivores that eat animal flesh. Bile is involved in the digestion of complex foods, in particular those containing high amounts of fat. This viscous substance is the paramount secretion of the liver, which produces up to 800 mL per day. It is mainly composed of water (97–98%), but also contains bile salts (0.7%), bilirubin (0.2%), fats (0.51%; predominantly cholesterol, but also fatty acids and lecithin), and 200 meq/L of inorganic salts. A detailed bile composition, based on signal assignments for ^1^H-NMR spectra, was provided by Molinero and colleagues [24]. Bile pigments are waste products generated by the degradation of old red blood cells; they include bilirubin, a yellow dye, and biliverdin, an oxidized form of bilirubin with a green color. Indeed, bile is a yellowish fluid that aids digestion by acting as a surfactant, helping fat emulsion by forming micelles, due to the amphipathic nature of the bile salts, thus making it easier for pancreatic lipases to hydrolyze triglycerides; micelles are also water soluble, thus facilitating absorption by the small intestine.

In the past, the biliary microbiota was analyzed to determine normal versus pathogenic microorganisms, as the gallbladder was used as a reference organ to test the immunity generated by particular vaccinations. This approach was used by Nichols and Stimmel in 1923 [25] to analyze the level of immunity against the typhoid group of infections developed in vaccinated guinea pigs. The rationale behind this experiment was that healthy animals, displaying a good immune system, would keep their gallbladders free from the pathogen, even when injected with *Salmonella typhi*. This publication also confirmed what was just a theory at the time: that the immunity developed by the animals was variable, and directly depended on the number of microorganisms injected during the vaccination. Gallbladder disease was also associated with the development of allergies, and even of cancer, with the latter appearing to mainly affect elderly people, and being rare in young individuals [26].

It is clear that the presence of gallstones, blocking the normal flow of bile and resulting in stasis, facilitates gallbladder infection (Figure 1). Gallstones are classified, according to their composition, into three types—cholesterol stones, pigment stones and mixed stones [27]—although some authors believe that this is not an optimal classification, and needs to be revised [28]. Pigment stones contain bilirubin and a low percentage of cholesterol, ranging from 10 to 20% (Figure 2 and Figure 3).

The microbiology of the gallbladder and adjacent areas is yet to be fully resolved, despite the advancements accomplished in recent years. One such report, by Li and co-workers in 2022 [29], used next-generation sequencing to identify the microorganisms present in the bile of patients with recurrent choledocholithiasis, and determine the key pathogens involved. The authors described that people suffering from the recurrent disease displayed a biliary microbiota containing higher relative amounts of *Fusobacterium* and *Neisseria*, as compared to stable patients lacking *Lactobacillus* species as well. In addition, they established a clear mutualistic relationship between *Fusobacterium* and *Neisseria*.

## 3. Microorganisms Colonizing the Gallbladder, Bile and Intestinal Lumen

Bile is continuously produced by the liver and reaches the gallbladder via the cystic duct, being discharged into the duodenum by means of the common bile duct (Figure 1). The sphincter of Oddi (also known as the hepatopancreatic sphincter or Glisson’s sphincter) is a muscled valve that opens and closes the duct to allow the flow of bile into the small intestine. As indicated above, gallstones can develop in the gallbladder and not produce any noticeable symptoms for years, but they can also result in gallbladder inflammation or cholecystitis. This inflammation is often caused by microorganisms originating from the small intestine, moved along either by peristaltic movements or by their own appendages. There are two important factors to take into account when studying microbial colonization: the “microbial residence time” (MRT), that is, the average amount of time that a given microorganism remains in a particular environment, and the “doubling time”, or the median time required for a microbial cell to divide (the doubling time can vary considerably from one bacterial species to another, but in the case of *E. coli*, a member of the family *Enterobacteriaceae*, it can be as low as 30 min at a temperature of 37 °C). In healthy humans, the gallbladder starts releasing bile around 30 min after a fat-rich meal, requiring approximately 1 h of rhythmic contractions for the organ to empty its contents into the intestine. The contractions are induced by a peptidic hormone (cholecystokinin, CCK or pancreozymin [30], produced by jejunal neuroendocrine cells [31], and encoded by a gene located on the short arm of chromosome 3 in humans (band 3p22.1). The hormone binds to its receptor, belonging to class A guanine-binding protein-coupled receptors, and induces the contraction of muscles in the gallbladder. Occlusions in the duct increase the bacterial time of residence, although not necessarily the doubling time, leading to a rapid accumulation of bacteria in the gallbladder, and producing inflammation. There are only two possible outcomes to resolve this condition: either the gallstone manages to pass through into the duodenum, or it has to be removed by surgery.

The harmful effects on human health of a bacterial buildup in an occluded gallbladder depend on the bacterial species colonizing it, and especially on the particular strain(s) involved. As is well known, some pathogenic bacteria can secrete toxins that can affect distal areas of the body. This is the case for *Salmonella* [32], in particular *Salmonella enterica*, a bacterium frequently encountered in gallbladders either containing calculi or suffering from inflammation [33]. *Salmonella* infection has an important role in gallbladder cancer [34], as the gallbladder is a reservoir for some species of this genus. *Salmonella* gallbladder infection increases secondary bile acid (SBA), leading to gallbladder carcinogenesis [35]. Gallbladder cancer is associated with the typhoid carrier state and positivity for the Vi antigen (a capsular antigen associated with *Salmonella*). This relatedness shows the relevance of the microbiome in the pathogenesis of multiple cancer types [36]. Pathogen species such as enterotoxigenic *Bacteroides fragilis* do indeed remodel the colonic microbiota, thus promoting colorectal cancer, possibly via IL-17 by T_H_17-cell-mediated inflammation. The process, however, could also begin in a microbiologically independent manner, but in the end be suppressed by the existing beneficial commensal microbiota [34].

The important matter of determining which microorganisms are present in a healthy human gallbladder was explored by Molinero and colleagues in 2019 [24] by analyzing microbial samples from patients suffering from lithiasis, which were obtained during gallbladder surgery, and comparing their microbiota to that colonizing healthy volunteers. Molinero and co-workers identified the microbial species colonizing the gallbladder by sequencing the 16S rRNA gene and are responsible for the first description of the “bile microbiota of healthy individuals”. According to these authors, the main bacteria present in bile belong to the *Firmicutes*, *Bacteroidetes*, *Actinobacteria*, and *Proteobacteria* phyla. Comparisons between microbes found in healthy and sick individuals indicated that, while members of the *Propionibacteriaceae* family were more abundant in the bile of healthy people, individuals suffering from gallbladder illnesses more frequently harbored bacteria belonging the *Bacteroidaceae*, *Prevotellaceae*, *Porphyromonadaceae*, and *Veillonellaceae* families (Table 1), with *Bacteroides* as the most abundant genus [37]. Microbial members of additional phyla, such as *Verrucomicrobia*, *Chlamydiae*, *Acidobacteria*, *Planctomycetes*, *Cyanobacteria*, *Spirochaetes*, and *Fusobacteria*, were also detected in the bile samples studied, although at low percentages [24]. An advantage of these DNA-sequencing studies is that the microorganisms do not need to be cultured in the lab, thus facilitating the analyses and allowing the detection of the non-cultivable microbial population. Unfortunately, there are still microorganisms that scientists have not determined how to grow under laboratory conditions, in addition to the fact that some organisms have a lower doubling time than others and, hence, their numbers in the bile samples would be overestimated when a cultivation step is included. The presence of members of the *Enterobacteriaceae* family in the bile of patients suffering from acute cholecystitis is not unusual, as these microorganisms are common inhabitants of the human GI tract [38], and bacteria colonizing the digestive system (mainly those present in the duodenum) can reach the gallbladder, provided that they are mobile and the sphincter of Oddi is malfunctioning. According to Chen and co-workers in 2019 [39], gallbladder dysbiosis is a major factor in the development of distal cholangiocarcinoma. However, the concept of gallbladder dysbiosis is not yet clear, as more information is required to definitely determine which microorganisms can be included in the “normal bile microbiota”. Saab and colleagues have recently characterized [40] the biliary microbiota in patients suffering from extrahepatic cholangiocarcinoma, as compared to healthy controls, by DNA extraction and 16S rRNA sequencing. These authors reported that, while there were no major differences between the two groups concerning *Proteobacteria*, the levels of *Bacteroidetes* were higher in cancer patients, and the opposite was true for the *Firmicutes* population. The authors identified the genera *Enterococcus*, *Streptococcus*, *Bacteroides*, *Klebsiella*, and *Pyramidobacter* as the prominent bacteria present in people suffering from cholangiocarcinoma.

Bacterial colonization of the gallbladder depends, to a great extent, on the type of bacteria involved, as Gram-positive and Gram-negative microorganisms behave differently in the presence of bile salts. The liver synthesizes a variety of acids [41], with the “primary acids”, cholic and chenodeoxycholic, as the most important in bile (Figure 4). These compounds, once conjugated with either taurine or glycine, generate the bile salts [42] that accumulate in the gallbladder and represent almost 80% of all the organic compounds present in bile. When the bile salts reach the intestine, their primary acids are partially dehydroxylated and the taurine and glycine groups are removed by the gut microbiota, producing the secondary bile acids, deoxycholic and lithocholic. An approximated 90% of these acids are recycled in the ileum and transported back to the liver and gallbladder, but it is important to note that, under normal conditions, they never reach the colon [43]. When either surgery or illness (for example, surgical ileum removal or Crohn’s disease) prevent the proper recycling of these compounds, the accumulation of bile acids in the gut can induce colorectal cancer [44,45]. Bile salts play a major role regulating the gut microbiota, thus contributing to intestinal homeostasis; these salts have been long known to possess antibacterial activity, through a variety of mechanisms including bacterial DNA damage and cell membrane disruption, thus playing a role in regulating the composition of the intestinal microbiota. Even more importantly, bile salts can control the expression of eukaryotic genes involved in the development of immunity [46]. In 2014, bile salts were reported to produce widespread protein unfolding (in vivo disulfide stress) as well as the aggregation of cytosolic proteins in bacteria, in addition to facilitating the formation of oxidized glutathione [47], thereby rendering many bacterial enzymes inoperative and preventing microbial growth. However, particular bacteria can overcome the effects of bile, with *E. coli* as the paramount example; in fact, bile salts are a key ingredient in the selective MacConkey medium, which allows the growth of a variety of Gram-negative bacteria, while inhibiting the growth of Gram-positive microorganisms. The generally accepted explanation for this phenomenon is that the presence of an outer membrane in Gram-negative bacteria acts as a barrier to bile salts [48].

In addition, bile salts, both cholate and deoxycholate, cause the aggregation of at least 83 cytosolic proteins, including components of both the 30S and 50S ribosomal subunits, enzymes involved in the transcription processes such as ATP-dependent RNA helicase, σ initiation factor and ρ termination factor, and the β and β′subunits of the DNA-dependent RNA polymerase, factors involved in the translation process such as the initiation factors IF-2 and IF3, and the elongation factors EFTu-2 and the EFG. Heat shock protein 33 (Hsp 33), encoded by the *hslO* gene, is a chaperone induced under oxidative stress conditions that prevents the aggregation of unfolded proteins [49]. Cremers and colleagues in 2014 [47] generated temperature-sensitive *Vibrio cholerae* strains and demonstrated that, when the bacterium lacks an active Hsp 33 protein, it becomes sensitive to both cholate and deoxycholate. Presumably, other Gram-negative bacteria that are resistant to bile salts could also have their resistant status overturned when lacking chaperone Hsp33, and this concept could also be extended to pathogens such as *V. cholerae*, a bacterium that requires the elongation factor Tu for protection against oxidative conditions [50].

Many Gram-negative pathogenic bacteria that invade the human gut and cause disease have to overcome a variety of host defense mechanisms, in addition to the human immune system [51]. These challenges include the acidity of the stomach and scarce iron availability [52], as well of a multitude of bacteriophages, the viral predators of bacteria, that are present in the human GI, some of which can even totally eliminate the pathogen. As indicated above, Gram-negative bacteria display a “natural” resistance to bile salts. As a matter of fact, some pathogenic microorganisms can use the bile compounds to enhance infection, hence under these conditions these biliary secretions are detrimental to the host that secretes them, as they can induce a variety of inappropriate immune responses. This is the basis of the concept denominated “survival of the fittest” [53], in this case applied to the ability of a bacterial pathogen to thwart the growth of symbiotic gut microbiota. In addition, several enteropathogenic bacteria use bile salt components to trigger and/or regulate virulence factors, allowing the microorganism to maintain infection (Table 2).

Some archetypal Gram-negative enteropathogens, such as *V. cholerae*, *Salmonella*, *Shigella*, and *Pseudomonas aeruginosa* (although this bacterium preferentially infects the urinary tract, it occasionally reaches the gastrointestinal tract, as it is resistant to a variety of compounds, including bile salts and brilliant green [54,71]) exploit this mechanism to assist in pathogenicity. Conversely, certain Gram-positive bacteria, including *Staphylococcus*, *Listeria monocytogenes* and *Clostridium perfringens*, are known to colonize the human gut, hence posing the question: how can they overcome the challenge posed by bile salts? In fact, even early publications described the presence of *Staphylococcus* in the bile of up to 43% of the patients [72], although they are known to be sensitive to cholic acid and its derivatives; this is also the case for other Gram-positive cocci, such as species of *Streptococcus*. Similarly, the sensitivity of *Streptococcus pyogenes* (group A, β-hemolytic) to sodium deoxycholate has long been used to describe the effect of cholic and deoxycholate on the cell walls of Gram-positive bacteria that results in cell lysis [73]. However, not all species related to the genus *Streptococcus* are lysed by bile salts, as in the case of *Enterococcus*, formerly known as Group D *Streptococcus* [74], which is a catalase-negative, non-motile bacterial genus that produces α-, β-, or γ-hemolysis on blood agar; this microorganism can readily grow in the presence of up to 40% bile and constitutes a significant pathogen, responsible for a high number of cases of endocarditis and intravascular infections, and has the ability to colonize the human gut and eventually reach the gallbladder. Other examples include members of the *Bacillaceae* family, such as *Bacillus* and *Clostridium*, which are sensitive to human bile salts only during vegetative growth, while their spores are totally unaffected by them.

The topic of variations in the human gut microflora has been revisited by microbiologists a number of times during the course of the last 100 years, frequently leading to different outcomes, mainly depending on the research approaches utilized. Because it is easier to study the gut microbiota composition and its changes in germ-free animals, most of the research has been carried out on rats that lack all microorganisms. Using this approach, Demarne and co-workers established in 1982 [75] that fecal lipids were mainly directly excreted as free fatty acids, triglycerides and, most importantly, as insoluble compounds in organic solvents. Already in 1965, Graber and colleagues [76] had reported that high-fat-containing diets altered the gut microflora, consistently generating an abundance of *Clostridium perfringens*, thereby providing the first evidence of the influence of the diet on bile salt production and cholesterol metabolism. On the other hand, this type of diet did not appear to affect bacteria such as enterococci, *Proteus*, lactobacilli, *Escherichia coli*, and *Staphylococcus aureus*, as compared to controls. A few years later, Drasar and co-workers reported in 1973 [77] changes in the human gut microflora in people according to their diets, in studies including a variety of dietary regimes such as a diet based on boiled mashed banana and matoke (Uganda), a diet high in rice (India and Japan) or the traditional Western diet, high in fat and animal protein content. The authors concluded that, while bacteria such as *Bacteroides* increased in protein-rich diets, dietary regimes high in carbohydrates benefited enterococci. In the matter of the amounts of bile salts entering the colon through Oddi’s sphincter [78], the research demonstrated that the concentration of bile salts was far higher in Western populations than in peoples residing in Uganda, India or Japan.

Germ-free chickens were also generated and used to study the proteases produced by the gut microbiota. This research revealed that the proteolytic enzymes did not appear to be involved in the degradation of the dietary proteins, but they played an important role in the lysis of endogenous proteins secreted into the intestinal lumen of the animals [79]. This fact, in addition, can have a major effect on the development of allergies, or other syndromes that influence the immunological status, if immunity-related proteins such as immunoglobulins type A are degraded.

The GI microbiota contains beneficial microorganisms, such as lactobacilli and Bifidobacteria, as well as harmful bacteria, including members of the microbial groups *Vibrionaceae*, *Pseudomonas aeruginosa*, staphylococci, clostridia (involved in toxin production, as well as either constipation or fatal diarrheas and the generation of potential carcinogens), and sulfate-reducing bacteria that generate toxic hydrogen sulfide (H_2_S). Mutualistic bacteria constitute a third group of microorganisms that are normal inhabitants of the GI tract, including *E. coli*, members of methanobacteria and *Veilonellaceae* groups, and anaerobic Gram-positive cocci that, in rare occasions, can colonize the gallbladder causing serious diseases [79,80]. A variety of publications has reported that fructo-oligosaccharides, together with other compounds considered as prebiotics, are specifically fermented by gut bifidobacteria. In these cases, those particular bacteria are the dominant species found in feces, which supports the current theory that the gut microbiota can be altered by manipulating the food composition of the diet.

## 4. Gallbladder and Infectious Diseases

The bacterium responsible for the infectious disease known as cholera (*Vibrio cholera*) [26] appears to use the gallbladder as a reservoir, and this pathogen can be isolated from hepatobiliary tract infections, even in asymptomatic long-term carriers [81]. Mayo Robson, in 1909 [82], provided a good analysis and classification of the different diseases affecting the gallbladder and bile ducts. Since microbial gallbladder colonization may be involved in gut dysbiosis, and hence have an effect on allergic reactions, it is worth summarizing the main pathological situations affecting this organ.

Cholecystitis is the main gallbladder affection, originally attributed to a *Bacterium coli commune* [83] (currently known as *Escherichia coli*, the most common bacterium in the patient’s colon), and normally generated by a blockage in the bile duct, due to a gallbladder stone. The occlusion produces acute cholangitis (inflammation of the bile duct, as a result of stasis and infection [84] that, in severe cases, can cause thickening of the wall of the gallbladder, making it very difficult to diagnose the disease by ultrasound examination. Cholecystitis symptoms include fever, cholic and abdominal pain, chills, and sometimes jaundice. The illness can progress to suppurative cholecystitis, with purulent material filling the gallbladder due to bacterial infection; this constitutes a serious health threat, which can result in death, if left untreated. Moffitt published an article in 1905 [85] that gives a comprehensible and straightforward description of the symptoms involved in gallbladder and bile duct infections; the author emphasizes the features of “colic”, a strong abdominal pain that can obscure prognosis and complicate diagnosis. An estimated 90% of gallbladder affections are related to the presence of gallstones (cholelithiasis), causing an apparent swelling of the gallbladder that is usually the result of a large stone completely preventing the biliary secretions from exiting into the duodenum. This blockage can generate epigastric pain around the navel area, but most often the ache originates in the gallbladder region and radiates upwards into the right subscapular region. When pain radiates to the left side of the body, it usually means that the pancreas is affected; this constitutes a serious health threat, as pancreatic necrosis is a life-threatening condition [86]. Often, the feeling of pain can spread downwards, along the colon and into the lower right quadrant of the abdomen, where the appendix is located, which can be wrongly diagnosed as appendicitis. Even the early publications recognized a link between the diet and biliary cholic, as well as establishing a connection between food and migraine attacks. The general advice to patients was not to ingest sugar, milk or eggs, warning them that they may feel hungry just before symptoms appear. Differential diagnosis can be facilitated by additional symptoms, including vomiting, stomach atony, hypo- or hyperacidity (which can occur in the same patient at different disease stages, even within the same day), and constipation. Evidently, the correct diagnosis is essential, as a variety of diseases, ranging from endocarditis to myocarditis and coronary sclerosis, can share some of the symptoms. The presence of jaundice can also help diagnose the disease, although it is not considered as relevant as initially believed; according to Moffitt in 1905 [85], 80% of patients with cholelithiasis do not display jaundice; hence, this symptom does not imply a gallbladder infection. On the other hand, the sonographic Murphy sign (also known as Sweeney’s sign), detected by palpation of the right subcostal area, was described to be 63% sensitive and have a specificity of 93.6%; hence, a positive Murphy’s sign is a good indication of the patient suffering from cholecystitis. This diagnostic maneuver is useful for identifying the location of the pain, in the right upper quadrant of the body, and was named after John Benjamin Murphy, an American surgeon. Murphy’s sign is negative in patients suffering from either choledocholithiasis (presence of gallstones) or pyelonephritis, kidney infection [87]. If bile samples are available, it is important to detect and identify any microorganism present in them, in order to determine whether the microbe is part of the patient’s “normal” microbiota (*E. coli*). Some early publications reported the presence of either *Bacillus aerogenes capsulatus* or *Bacillus welchii*, currently known as *Clostridium perfringens*, a capsulated Gram-positive, spore-forming bacterium [88], which is occasionally involved in gallbladder necrosis [89].

Gallbladder infections can also produce bleeding that should not be disregarded, as it can be severe [90,91]. Even in the early stages of gallbladder surgery, some of the surgeons proposed that the microorganisms infecting this organ and its ducts reached the gallbladder via the blood stream, or even the lymphatic vessels [92]. It is important to note that many, if not all, of the bacteria present in the gallbladder are motile; hence, they can travel into the gallbladder ducts from the duodenum, where the bile empties into the intestine. According to Judd in 1921 [92], bacteria colonizing the gallbladder can reach the pancreas through the lymphatic system, resulting in an inflammation known as pancreatitis; this was later confirmed by Weiner and colleagues in 1970 [93]. Although, according to some sources, gallbladder infections originate from appendicitis, those infections can also be suffered by people who have already undergone an appendectomy. It is interesting to note that gallbladder disease resulting from the uncontrolled growth of microbial pathogens mainly occurs in patients that also suffer from arterial hypertension, arteriosclerosis and myocarditis [94].

## 5. Gut Microorganisms and Allergies

The association between intestinal bacteria and allergic diseases was recently reviewed by Han and co-workers in 2021 [95]. Food allergies, hay fever, anaphylaxis, atopic dermatitis, and asthma are mainly due to the anomalous synthesis of immunoglobulin E (IgE). Hay fever, in particular, was the first allergy recognized, and described as such, in 1819, by John Bostock. In 2011, Ramachandran and Aronson [96] reported Bostock’s discoveries, in particular the clinical case he presented to the Medical and Chirurgical Society on the 16 of March 1819: “Case of a periodical affection of the eyes and chest, the first recorded description of what he later called “catarrhus aestivus” or summer catarrh, and which soon became known as hay fever”.

This type of symptomatology is only possible with the participation of IgE molecules, which are monomeric antibodies with a molecular mass of about 190 kDa and other characteristics originally described by IshizakaK and colleagues in 1966 [97]. IgE is the main participant involved in anaphylactic responses, as well as in hay fever and immunological responses to parasites (Figure 5) [98]. Although some of these allergies can be prevented by exposure to potential allergenic compounds early in life, allergen sensitivity can result in the development of allergic diseases in susceptible individuals. Immunotherapy with allergens, also known as desensitization or hypo-sensitization, has already been described [99]. These treatments can involve, for instance, a medical treatment against asthma, consisting of patients being exposed to progressively larger amounts of the allergen responsible for the condition, resulting in the alleviation, or even the cure, of the disease [100,101].

The main reason(s) why different individuals react differently to a given allergen, in ways that can even result in an allergic disease, is yet unknown. Nevertheless, it is generally accepted that the individual’s genetic background must play a major role, along with factors such as the diet and the composition of the gut microbiota [102]. Recent studies suggest that the current high incidence of food allergies is most likely due to the changes in the GI microbial populations generated by our “modern eating habits”, particularly in developed nations. In fact, the microorganisms that inhabit our bodies, our gut microbiota, are now envisioned as “a second brain”, that uses chemical mediators (often generated by the gut microbiota) to modulate distal organs, even having an effect on our mood and behavior [2]. The review article by Ipci and co-workers in 2017 [103] provides a currently accepted view on the possible mechanisms involved in the trigger and modulation of allergic diseases by the microbiota. The gut microbiota influences systemic immunity through the secretion of IgA, and these immunoglobulins can, in turn, affect the microbial growth (bacteria, fungi, protozoa, or viruses); actually, they can influence the microbiota present in the oropharyngeal area and even the lungs. Gut dysbiosis, caused by a variety of reasons including oral antibiotic therapy, can drastically affect human health (Figure 6), even potentially causing death, as can be the case for pathogens such as *C. difficile*. Examples of other immune diseases, including autoimmune disorders, either believed or confirmed to be directly associated with alterations in the composition of the gut microbiota are described in the following sections.

### 5.1. Systemic Lupus Erythematosus

Lupus (from the Latin word that means “wolf”), or systemic lupus erythematosus (SLE), is an autoimmune disease (believed to be a type III hypersensitivity) in which the patient’s immune system does not recognize different organs of the body as part of the “self” [105,106].

The condition is believed to have a genetic background, because studies with identical twins demonstrated that if one of the individuals suffers from the disease, then the twin sibling has a 25% probability of also developing the syndrome, although a risk factor as low as 1:4 indicates that there must be other factors involved in the development of the disease. In recent years, the gut microbiota has been proposed as a major player in the illness development mechanism, as the diet can either ameliorate or aggravate the condition. Current data support an association between the microbiota composition and SLE, with reports indicating that the GI microbiota of lupus patients contains reduced quantities of Firmicutes/Bacteroides [107] in comparison to healthy individuals. For their part, Wen and colleagues described in 2021 [108] an increase in *Proteobacteria* and *Enterobacteriales*, as well as a decrease in the number of *Ruminococcaceae* bacteria present in the GI microbiota of lupus patients; the authors also reported that some metabolites, amino acids and short-chain fatty acids were also enriched in people suffering from the disease. However, these results are far from definitive as, according to Ruiz and colleagues in 2018 [109] and Yacoub and co-workers in the same year [110], considerably more research is still needed in this field but, even at this stage, new applications are emerging for the treatment of this devastating disease. New therapies include fecal matter transplantation (FMT), live biotherapeutics and alterations in dietary intake, all of which can ameliorate symptoms and are becoming an important part of the treatment of lupus [104]. Indeed, as indicated above, gut dysbiosis can lead to the development of severe infections, which have a significant effect on both morbidity and mortality in Lupus patients. Another major complication of lupus is what is commonly known as a “leaky gut”, referring to intestinal leakages into the blood stream, possibly due to malfunctioning of two proteins involved in the formation of tight junctions (claudin and occludin), which eventually lead to the loss of function of the intestinal barrier, allowing bacteria and/or their toxins to reach distal body organs. Some of the molecules produced by the pathogens infecting host tissues or organs can contain antigen structures that are similar to compounds produced by human cells, a phenomenon known as “molecular mimicry”. Once the complement is activated at the C3 level, it can attack self-antigens that resemble bacterial components, resulting in tissue lysis and the onset of an autoimmune response mainly mediated by T-helper 17 (Th17) cells [111].

### 5.2. Leaky Gut Syndrome

From the microbiological point of view, as indicated above, the gut is an open system with two ends accessible to the environment, where a vast variety of microorganisms and viruses resides. This open ecosystem must be separated from the circulatory system by a delicate yet complex structure, the intestinal epithelium, which uses tight junctions to prevent interactions between the two physiological compartments. These junctions are mainly maintained closed by two cooperating molecules: claudins (a family of small 20–27 kDa transmembrane proteins [112,113], and occludin (a 65 kDa plasma membrane protein that oxidizes NADH [114]. The proper interaction between claudins and occludins plays a major role in homeostasis. Any deviations, either in the intestinal barrier permeability or in the associated lymphoid tissues, can result in the so-called “leaky gut syndrome”, leading to the potential development of various autoimmune diseases.

Tight junctions are in turn regulated by another protein called zonulin, a haptoglobin 2 precursor that represents the mammalian counterpart of the zonula occludens toxin, secreted by *Vibrio cholerae* [115,116], which has been reported to be involved in both Crohn’s disease and ulcerative colitis, as well as in preventing colonization by intestinal microorganisms [117]. Malfunctions in the regulation exerted by zonulin result in the development of leaky gut syndrome. Consistently, zonulin was found to be overexpressed in patients suffering from autoimmune diseases. According to Alessio Fasano in 2012 [118], the epithelial paracellular space is estimated to measure 10–15 Å, which means that, under normal conditions, any molecule with a molecular mass exceeding 3.5 kDa cannot travel across the epithelial tight junctions. Zonulin has been identified as a haptoglobin precursor [119], and its mature form is composed by both α- and β-polypeptide chains; with the α-chains displaying differences in their molecular mass (9 or 18 kDa, for α1 and α2, respectively), while the β-chains have a mass of 36 kDa. So, in turn, if involving occludins, then the appropriate term to use would be “zonula occludens”, but there are two more cases in which the integrity of the intestinal wall would be at stake: (i) “zonula adherens”, also known as “adherens junctions” or “intermediate junctions” formed by protein complexes to be found in cell–cell junctions, cell–matrix junctions in epithelial and endothelial tissues, and that are comprised of α-catenin, β-catenin, and E-cadherin. In fact, dysbiotic microbiota including *Helicobacter pylori* may induce a breakdown of the tight junctions through virulence factors such as CagA in this pathogenic bacterium; and (ii) “desmosomes”, also known as “macula adherens”, located on the lateral side of the plasma membranes, which exhibit very strong cell-to-cell adhesion, and are hence located in tissues with much mechanical stress, such as the cardiac muscle. As indicated by Camilleri in 2019, desmosomes in particular, which are positioned beneath the apical complex, are indeed quite complex structures with desmoglein, desmocollin, desmoplakin and keratin filaments performing interactions [120,121,122,123,124].

There are several human autoimmune diseases that involve zonulin, including celiac disease (triggered by the presence of the wheat protein gliadin in the diet of susceptible people [125,126,127,128], and type 1 diabetes, an illness where an increase in intestinal permeability precedes pancreatic Langerhans isle destruction [125]. Indeed, Watts and colleagues confirmed in 2005 [129] that zonulin was involved in the pathogenesis of type 1 diabetes (at least in rat models) and reported that the increase in intestinal permeability occurred two to three weeks prior to the actual onset of the disease. Other autoimmune diseases so far described to involve zonulin, as well as an increase in intestinal permeability, are Crohn’s disease and ulcerative colitis [130], which display observable changes in the expression of claudins [131]. Additional illnesses include asthma [132,133], and autoimmune disorders such as multiple sclerosis [118] and ankylosing spondylitis [134]. A recent review by Gierynska and co-workers in 2022 summarizes the importance of maintaining the integrity of the gastrointestinal epithelium and the mutual relationship with the gut microbiota [135]. So, molecules with a molecular mass higher than 600 Da are readily absorbed in cases of inflammation or when the tight junctions become loose. Additionally, the pore pathway by which water or ions enter into the cells may be affected by interleukin-13 in the opening, and by tumor necrosis factor (TNF-α) in cases of leaky situations; their modes of action may be found in Usuda and colleagues’ paper of 2021 [136].

### 5.3. Polyamines and Food Allergies

Polyamines are organic compounds containing two or more amino groups that can either be produced naturally or synthesized in vitro. The first class includes diamines such as putrescine and cadaverine (cadaverine and histamine are also biogenic amines, but they are considered polyamines), the triamine spermidine and the tetra-amine spermine. Putrescine is present in all organisms [137] and plays a number of different roles, including the stabilization of DNA and repair of double-stranded DNA breaks, functioning as an antioxidant and in cation substitution, as well as interacting with mitochondria and chloroplasts in plants, and being involved in cellular transport. This compound, together with cadaverine, was originally described by Ludwig Brieger in 1885 [138], and can be chemically synthesized, either by the hydrogenation of succinonitrile or by a biotechnological approach. Qian and co-workers [139] reported the construction of a metabolically engineered *E. coli* strain, containing a recombinant ornithine decarboxylase (EC 4.1.1.17) gene, which could synthesize putrescine (a compound containing four carbons) from ornithine, with a yield reaching 0.75 g/L /h. As ornithine decarboxylase is essential to promoting cell growth, its inactivation or deletion triggers cellular apoptosis. The regulated degradation of ornithine is mediated by the cellular proteasome in an ubiquitin-independent manner [140]. Cadaverine is a 5-carbon molecule, which can also be produced by recombinant *E. coli* cells containing L-lysine decarboxylase (EC 4.1.1.18), an enzyme that directly converts L-lysine to cadaverine with a yield of 0.32 g/L/h [141]. Spermidine, and then spermine are produced by two successive aminations of the propylamine group of adenosylmethioninamine, catalyzed by spermidine synthetase (EC 2.5.1.16) [141,142]. The decarboxylation of L-arginine (EC 4.1.1.19) results in the generation of an additional polyamine (agmatine), as reported by Moinard and colleagues in 2005 [143], although agmatine was originally discovered by Albrecht Kossel in 1910 [144]. Polyamine interconversions occur via a chemical process known as acetylation, which utilizes acetyl coenzyme A as a donor in a procedure that involves the enzyme polyamine N′-acetyl-transferase (EC 2.3.1.57), followed by cleavage with polyamine oxidase. Examples of this process include the synthesis of spermidine from spermine, and that of putrescine from spermidine. In humans, the first enzyme is encoded by the *SAT1* gene, located on the X chromosome, while the second step is carried out by a flavoprotein, generating three compounds (hydrogen peroxide, an amino-aldehyde and a primary amine, which are assigned different EC numbers according to their eukaryotic or prokaryotic origins: EC 1.5.3.13, EC 1.5.3.14, EC 1.5.3.15, EC 1.5.3.16, and EC 1.5.3.17).

All natural polyamines exhibit low oral toxicity. It is thought that an insufficient amount or the absence of polyamines in the dietary intake can cause allergen sensitization, which in turn can lead to specific food allergies [145]. Indeed, the probability of a particular infant developing a food allergy can be as high as 80% when the spermine content of the milk used to feed the baby is below 2 nmol/mL The amount of putrescine, spermine and spermidine present in the fruits and vegetables normally included in the human diet varies considerably; for instance, oranges contain 1330, 13 and 8 units (1 unit = 1 μg/100 g) of putrescine, spermidine and spermine, respectively, while grapes only contain 9, 6 and 2 units of the above polyamines, respectively. A publication by Larqué and colleagues [146] describes the polyamine content of a wide variety of human foods. Diet is the main source of polyamines, but the body can also endogenously produce polyamines from pancreatic secretions and the intestinal luminal bacteria [147], as well as from exfoliated enterocytes [148]. As reported by Larqué and co-workers, the polyamines supplied in the diet are absorbed in the duodenum and jejunum by a mechanism of passive diffusion and distributed to distal organs and tissues throughout the body, where they play a role in cellular growth. In addition, they are uptaken in the gut mucosa by both normal and neoplastic epithelial cells, in what appears to be an active transport process [149]. Putrescine is transported by a sodium-independent, although time- and temperature-dependent, mechanism characterized by a Michaelis constant (Km) of 1.26 × 10^−6^ M, and a maximal velocity of reaction of 5184 pmol putrescine/mg protein/h [150]. Scemama and colleagues characterized, in 1993 [151], univectorial polyamine transport systems in duodenal cells, and determined their Michaelis constants.

The concentration of polyamines in humans tends to decrease with age, but can be replenished, at least to a certain extent, by the secretions of intestinal bacteria. It appears that food supplements containing polyamines could mitigate the organ deterioration associated with age, including in the brain. Kibe and colleagues reported in 2014 [152] that the inclusion of arginine in the diet of mice produces high levels of putrescine in the colon, as well as spermine and spermidine in the blood, in addition to an increase in bifidobacteria, resulting in the suppression of inflammation, as well as a significant increase in animal longevity as long as the polyamine concentration in the body was kept high. In summary, diet supplementation should, at least in principle, result in lower levels of inflammation as well as the amelioration of food allergies. Whether this positive response can also be triggered in humans is yet to be determined, although early indications suggest that these compounds can also have a favorable effect, increasing longevity, as indicated by Eisenberg and colleagues in 2009 [153], but this matter needs further investigation. Indeed, these authors are optimistic about the role of polyamides increasing the lifespans in a variety of cells and organisms, including yeast, flies, worms, and human immune cells. It is currently believed that the mechanism involves epigenetic effects that could involve deacetylation of the H3 histones.

Buts and co-workers described in 1993 [154] that spermine is involved in the maturation of the small intestinal microvilli, as determined by the increase in microvillus marker enzymes such as lactase, β-fructofuranosidase and aminopeptidase, and that the process is specific to spermine. To the best of our knowledge, this represents one of the first experimentally demonstrated effects of a classic polyamine in intestinal homeostasis. This mechanism includes the regulation of the synthesis of occludin by intestinal epithelial cells, as later demonstrated by Guo and colleagues in 2003 [155]. The psychological process of memory can also be affected by the concentration of polyamines, and these molecules can alleviate the signs and symptoms produced by certain syndromes, such as Huntington’s disease, a progressive neurodegenerative disorder associated with motor and cognitive impairment. In a rat model of this illness, Velloso and colleagues [156] demonstrated that the induction of a Huntington-like disease by quinolic acid could be reversed, at least partially, by a single injection of spermine (0.1 or 1 nmol/site).

On the topic of inflammatory reactions, Zhang and co-workers published in 1997 [157] that, in experiments conducted with mononuclear cells activated by treatment with bacterial lipopolysaccharide (LPS), exogenously added spermine downregulated the synthesis of active cytokines, including interleukins (IL-1 and IL-6) and macrophage inflammatory proteins (MIP-1α and MIP-1β) as well as tumor necrosis factor (TNF). Cytokine downregulation was not just triggered by LPS stimulation, but the activation process could also be completely reversed. The function of polyamines in maintaining the intestinal integrity was demonstrated by Li and co-workers in 2001 [158], by showing that polyamine depletion resulted in the activation of the inflammation-associated NF-κB transcription factor along with increased cellular apoptosis.

Unfortunately, the dietary addition of exogenous polyamines can have negative health effects, since Gram-negative bacteria with fermentative metabolism, such as *E. coli*, in the presence of particular polyamines (such as putrescine) can induce a specific catabolite pathway that degrades ϒ-aminobutyrate into succinate, as the polyamine is used as a nutrient source. This process can cause a general decrease in intestinal polyamines, hence negatively affecting the gut microbiota [159].

### 5.4. The Gut Microbiota Can Prevent Food Allergies and Help Maintain Oral Tolerance

It is generally accepted that the host can perceive the microorganisms present in its gut microbiota by a variety of sensors, including toll-like receptors (TLRs) such as TLR4 for commensal flora, as well as nucleotide-binding oligomerization domain (NOD)-like receptors [160]. In this regard, saprophytic bacteria do not usually elicit any immunological responses, although they can enter lymphatic nodes, while pathogenic species, such as *Salmonella typhi*, can disturb gut homeostasis and colonize the blood stream, even reaching distal body organs. Currently, food allergies probably constitute the main disorders affecting the infant population. This can be due to a variety of reasons including stress, bottle-feeding instead of breastfeeding with processed and pasteurized cow’s milk, and cesarean delivery as opposed to vaginal delivery. Moreover, it is mainly prevalent in industrialized countries where the indiscriminate use of antibiotics is extensive in the pharmaceutical and food industries [161,162]. Statistically, the number of people affected is exponentially increasing, and the condition escalates the burden on hospital services, as well as expanding the cost of health care. As is well known, food allergy is the result of an immune system overreaction by a Th2 pathway, with the patient’s body responding to innocuous antigens contained in the diet [163]. In fact, it is a failure of the mechanism known as “immune tolerance”, involving Treg cells. These authors reported that in ovalbumin (OVA)-sensitized Il4raF709 mice (mice with an allergic response to eggs), ovalbumin resulted in specific changes in the gut microbiota of the animals, leading to high numbers of *Lachnospiraceae*, *Lactobacillaceae*, *Rikenellaceae*, and *Porphyromonadaceae*, a bacterial profile that was not present in wild-type mice lacking the allergic response to ovalbumin. In addition, Rivas and co-workers [164] reported that the allergic symptoms disappeared if the animals were treated with OVA-specific Treg cells, and most interestingly, when the bacterial microbiota in wild-type mice was replaced with the microorganisms present in the GI of OVA-sensitized individuals, the allergic response to ovalbumin was replicated in these previously healthy animals. These results are similar to data from previously published studies, in particular the research carried out by Kim and co-workers [28], who reported that oral probiotic bacteria (such as *Bifidobacterium bifidum* and *Lactobacillus casei*) could suppress the OVA-induced allergy in mice. Experiments performed by Sonoyama and co-workers [165] with BALB/c mice, an immunodeficient laboratory strain of the animals, revealed that feeding the animals a rice variety known as Yukihikari produced changes in their gut microbiota that prevented signs associated with food allergy, such as diarrhea. In addition to the usual GI microorganisms, belonging to the groups *Actinobacteria*, *Bacteroidetes*, *Firmicutes*, *δ-Proteobacteria*, *ε-Proteobacteria*, and *Verrucomicrobia*, the mice displayed high numbers of *Akkermansia muciniphila*, a mucin-degrading bacterium often present in the human gut [166,167] that belongs to the phylum *Verrucomicrobia*, and constitutes a bacterium closely related to *Verrucomicrobium spinosum* (with a similarity of 92%, based on the16S rRNA gene sequence analysis). The novel isolate is a Gram-negative, strictly anaerobic, non-motile, non-spore-forming microorganism, which produces a capsule when grown in a mucin-containing medium. This novel bacterium was proposed to play a role as “gatekeeper” of the human mucosa [168]. The amount of viable *Akkermansia muciniphila* cells in the gut mucosal layer was reported to decrease in both obese and type 2 diabetic mice, as compared to healthy animals. In addition, treatment with *A. muciniphila* reversed the above conditions, caused by either high-fat diets or insulin resistance [169]. It is currently accepted that there is a crosstalk between the gut microbiota and dietary lipids, aggravating white adipose tissue inflammation in a process involving TLRs and chemokine CCL2 [165]. CCL2 is a small chemokine, also known as “monocyte chemoattractant protein 1”, that is involved in monocyte recruitment and memory T-cells, as well as in the tight regulation of cellular mechanisms [170]. The gene encoding this signaling protein is located on chromosome 17 in humans (17q11.2–q21.1) [171], and there are two cell surface receptors, CCR2 and CCR4, that bind CCL2 [172].

Perhaps the main bacterial metabolite produced by the gut microbiota is butyrate, a molecule involved in immune tolerance, as well as displaying strong anti-inflammatory effects in a variety of allergic diseases. In infants, butyrate appears to be associated with triggering food immunotolerance, in particular during the first 1000 days of life [173,174]. In addition to butyrate, other short-chain fatty acids (SCFAs) produced in the human gut by bacterial fermentation, such as acetate, valerate and propionate, display multiple beneficial effects, such as a protective role in autoimmune and inflammatory reactions [175], that are achieved by modulating the metabolic status of T-cells via epigenetic changes. Additional compounds produced by the GI microbiota include aryl hydrocarbon receptor (AHR) ligands and polyamines. AHR is a ligand-activated transcription factor, produced in high amounts by enteric bacteria, involved in the recognition of indole and tryptophan catabolites, that often protects against the onset of food allergies [176,177,178,179]. The AHR receptor is also suspected to be involved in the development of gastric cancer triggered by *Helicobacter pylori* [180]. Polyamines, on the other hand, or at least milk polyamines, are reported to play preventive roles against the development of food allergies [181]. It is currently known that polyamines such as spermine and spermidine, in addition to displaying antioxidant properties, are involved in the development of the immune system, as well as in general cell proliferation [181,182].

As a matter of fact, many of the compounds mentioned above are described to either “alleviate stress induced by the bidirectional brain–gut axis alterations”, or display tumor-suppressing properties [183]. SCFAs have several receptors, including the G-protein-coupled receptors 41, 43 and 109, as well as the olfactory receptor 78, with the latter actually being expressed in intestinal epithelial cells, immune cells and adipocytes. Deficiencies in receptor-mediated signaling can lead to high blood pressure and cardiovascular diseases. Recently, “a unique gut microbiota signature” was associated with pulmonary arterial hypertension, a disease that can often be fatal. This “signature” includes bacteria such as *Anaerostipes rhamnosivorans*, *Amedibacterium intestinale* (formerly, *Eubacterium dolichum)*, *Ruminococcus bicirculans*, and *Ruminococcus albus* [184,185,186,187,188,189].

*Faecalibacterium prausnitzii* is a Gram-positive anaerobic (extremely oxygen-sensitive), non-spore-forming, non-motile rod, that was classified as part of a novel genus by Duncan and colleagues in 2002 [190]. This bacterium, previously considered as part of the *Fusobacterium* genus, is an inhabitant of the human gut, where it is the most abundant microorganism, as it can constitute up to 5% of the gut bacteria present in healthy adults that do not suffer from food allergies. *Faecalibacterium prausnitzii* was proposed as one of the best indicators of a healthy status in the human gut microbiota and it is believed to alleviate allergic reactions that may include food allergies. The *F. prausnitzii* population is drastically reduced in patients suffering from conditions including inflammatory bowel disease [191]. This bacterium is also considered a next-generation probiotic, with an application as a biotherapeutic agent to alleviate food allergies and restore a healthy gut microbiota. However, its putative use in therapy must be carefully assessed as, according to Sjödin and colleagues in 2016 [192], there is a strong association between high numbers of *F. prausnitzii* and eczema incidence. Codoñer and co-workers [193] reported that the overabundance of the bacterium, at values higher than 5%, was associated with the development of psoriasis, and on the other hand, lower microbial percentages were described to be associated with asthma, including mite-induced allergies [193,194,195], Crohn’s disease [196], major depressive disorders [197,198], and even obesity [199,200,201]. It appears that treatment with *F. prausnitzii* can improve certain conditions caused by microbial gut dysbiosis, but it is very challenging and cumbersome to grow *F. prausnitzii* in laboratory conditions due to its extraordinary sensitivity to oxygen. The difficulties described could be overcome by using the FMT technique to transfer feces from healthy donors who contain high levels of *F. prausnitzii* into the GI of people suffering from allergies [3]. This is an easy technique to perform, as the fecal matter can be transplanted either by an enema or through a nasogastric tube. In reality, FMT is not a new development as it was described in China more than 1000 years ago as a treatment for putative fatal watery diarrheas, for which local physicians used what was commonly denominated “yellow soup”, containing either fermented or fresh fecal products. FMT was applied by Björkqvis and co-workers [202] as a cure for recurrent *Clostridioides difficile* infections and involved the transplantation of feces from healthy patients that were rich in the bacterium *F. prausnitzii*.

In addition to butyrate, *F. prausnitzii* produces other compounds with beneficial effects on the host, such as salicylic acid, shikimic acid and raffinose. Salicylic acid can be used to generate mesalamine (5-aminosalicylic acid, 5-ASA), while shikimic acid is a precursor for the synthesis of several aromatic compounds that, as is the case for 5-ASA, reduce tissue/organ inflammation. These authors also reported the existence of seven peptides derived from a protein of 15 kDa known as MAM (microbial anti-inflammatory molecule), which displays anti-inflammatory properties [203] and is absent in patients suffering from Crohn’s disease. Sokol and colleagues had already reported in 2008 [204] that some of the metabolites produced by *F. prausnitzii* exhibit anti-inflammatory properties.

Some dietary compounds can induce a beneficial health effect by modulating the gut microbiota. This has been shown to be the case for inulin, a member of the fructan group of polysaccharides and synthesized by many plants as energy storage, because its presence in the diet was described to result in an increase in the numbers of bacteria such as *F. prausnitzii* and *Bifidobacterium adolescentis* [205]. In addition to what was mentioned above, *F. prausnitzii* promotes multiple beneficial effects and, consequently, appears to be a suitable treatment for diseases affecting the human gut, including food allergies. Accordingly, Rossi and colleagues described in 2015 [206,207] that the extracellular polymeric matrix of this bacterium can attenuate the clinical signs of some types of colitis, as well as, one year later, reporting that the bacterium has a strong ability to induce IL-10 and to modulate T-cell responses [202]. In fact, Carlson and co-workers had already established in 2013 [208] that this microorganism can potentiate the function of the intestinal barrier.

*Ruminococcus* is another bacterial genus that can be either beneficial or detrimental to human health. These microorganisms are part of the GI flora, and the genus contains species that are useful, while others are harmful. Berin [7], as well as Bao and colleagues [209], studied two sets of healthy identical twins, and provided two examples of bacterial species, *R. bromii* and *Phascolarctobacterium faecium*, that were clearly associated with a healthy fecal metabolome, probably protecting the GI from developing food allergies. *Ruminococcus flavefaciens* is an example of a GI-protective bacterium in mice, as it induces changes in both upstream (i.e., mitochondrial oxidative phosphorylation) and downstream pathways associated with cortical gene expression [210]. These authors also described the effect of antidepressants currently used to treat mental health conditions on altering the gut microbiota via the brain–gut axis. Other microbial species, however, are associated with allergic disease in both human infants and mice, with *Ruminococcus gnavus* being a prime example. This microorganism, known as an “unfriendly bacterium”, produces inflammation of the airways and increases the number of T-helper 2 (Th2) cells in both the colon and the lungs [211]. *Phascolarctobacterium faecium* is another example of a beneficial bacterium that was originally described by Dot and colleagues [212] in koala feces, and constitutes a Gram-negative, obligate anaerobic, non-sporulated rod that is phylogenetically related to the genera *Clostridium* and *Bacillus*. *P. faecium* ferments succinate, mainly producing propionate and acetate, SCFAs that, as indicated above, are known to strengthen the gut barrier function and alleviate food allergies. According to Wu and co-workers [213], *P. faecium* is present in healthy individuals at around the first birthday, increasing steadily, first in children and then in adults up to the age of 60, after which it gradually decreases.

In summary, as reported by Canani and colleagues [162], “*We are approaching a new era in which we can regulate immune system development and function through dietary intervention and measure the clinical impact through gut microbes and their metabolites*”.

## 6. Omic Approaches Applied to the Study of Gallbladder Metagenome, Proteome, Transcriptome and Metabolome

The human proteomes have been studied in both healthy and pathological cell types and in cell lines. Moreover, the transcriptome of a cell and/or tissue can also be determined and quantified by next-generation sequencing technology (RNA sequencing, RNA-Seq). A variety of cell types is located in the gallbladder, including glandular cells, fibroblasts, endothelial cells, smooth muscle cells (innervated by the appropriate nervous terminals of the sympathetic system that activate α_2_ adrenoreceptors located on vagal terminals in the gallbladder ganglia), inflammatory cells, and others. The characterization of this organ may be performed by antibody-based proteomics and transcriptomics of each cell type, with the final identification of its global protein profiles. An overall tissue protein analysis of the human gallbladder is therefore vital for the identification of molecular regulators as well as the effectors of the physiology involved [214]. These authors have contributed to the understanding of the gallbladder phenotype in healthy states and quantified the genes overrepresented in such an organ; by comparing to the transcriptome of 27 other human tissues, they identified no fewer than 140 gallbladder-specific proteins exhibiting elevated expression. Furthermore, they investigated within the tissues the cellular localization of the genes encoding these specific proteins by combining the use of antibody-based protein profiling and the integration of data from the Human Protein Atlas (HPA) portal, Uniprot, and the gene ontology cellular component (GO-CC) databases. In addition, they also explored the biological processes carried out by these proteins by using the GO biological process (GO-BP) and KEGG Pathways, and finally, they identified the metabolic reactions associated with them through the use of the human metabolic reaction database (HMR) 2.0. [209]. With all these approaches they demonstrated the usefulness of the combined analysis of transcriptomics and affinity proteomics. Nepal and colleagues analyzed in 2021 [215] the mutational landscape of gallbladder cancer, which is the most common tumor of the biliary tract. Whole-exome sequencing and targeted sequencing were the techniques employed in order to determine the transcriptomes and DNA methylomes. The results suggested that aflatoxin exposure was associated with the increased survival of these tumors. Several other studies were indeed involved in the identification of the critical genes that differentiate the normal gallbladder from the cancerous organ [216].

Next-generation sequencing offers advantages to the knowledge of the human microbiota by whole-metagenome shotgun (WMS) sequencing and 16S rRNA sequencing, and accordingly several studies have been elucidating the biliary microbiota. Shen and collaborators analyzed in 2015 [217] samples from 15 Chinese patients with gallstone disease in order to determine their microbiota. They identified no fewer than 13 novel biliary bacteria based on WMS sequencing, and the genes encoding putative proteins related to gallstone formation and the phenotype of microbial bile resistance (e.g., β-glucuronidase and multidrug efflux pumps). The results showed that the patients suffering from gallstones had reduced microbial diversity compared to healthy individuals.

Kose and co-workers analyzed in 2018 [218] the cultivable gallstone-associated bacteria and their 16S rRNA profiles, thus providing indirect evidence of the interaction between the processes involved in gallstone formation and the microbiota. As a matter of fact, the presence of pigmented stones revealed the prevalence of the genes so far involved in carbohydrate metabolism, while the cholesterol stones showed a genic profile altogether dominated by protein metabolism, possibly reflecting, according to these authors, the chemical differences in the biofilms originated by Gram-negative and Gram-positive bacteria. In a different study, the bile and gallbladder mucus microbiota were analyzed by different techniques. Culture-dependent techniques, 16S rRNA metagenomic data generation and proteomic analysis using an electrospray ionization (ESI) linear trap quadrupole (LTQ)-Orbitrap mass spectrometry (MS) were used to display the gall bladder microbial ecosystem, which was mainly populated by members of the phyla Proteobacteria, Firmicutes, and Bacteroidetes. Furthermore, fluorescent in situ hybridization (FISH) and transmission electron microscopy (TEM) allowed the visualization of bacteria of different morphological types [219]. A recent approach evaluated the different bile samples by lipidomics and metagenomics in order to perform an early detection of gallbladder cancer. The results showed that the increase in specific bacterial taxa (*Leptospira*, *Salmonella enterica*, *Mycoplasma gallisepticum*) correlates with specific lipids (ie. lysophosphatidylinositol, ceramide 1-phosphate, and lysophosphatidylethanolamine) and the development of gallbladder carcinoma [220]. In this regard, omics such as proteomics, transcriptomics and metagenomics, are important tools to obtain a global view of the gallbladder functions and interactions, and even understand the development of the related diseases.

In addition, many studies related to the gallbladder have been provided by metabolomics. Common methods used in metabolomics are mass spectrometry and nuclear magnetic resonance (NMR) spectroscopy [221]. NMR technology has been employed to analyze the lipid profiles of gallbladder tissue extracts, with the outcome that alterations in the lipid metabolism may be detected upon comparison between benign and malignant gallbladder diseases [222]. Additional research based on NMR analysis reported on the variations in the concentrations of low-density lipids (LDLs), very-low-density lipids (VLDLs), branched-chain amino acids (BCAAs), alanine, glutamine, creatinine, and tyrosine, 1,2-propanediol, pyruvate, glutamate, and formate in gallbladder cancer as compared to healthy individuals [223]. Moreover, gallbladder stones from patients from different geographical regions have been analyzed using solid-state NMR, showing different levels of cholesterol, bilirubin, and calcium carbonate correlations [224].

NMR technology has also provided useful information about differences in metabolite concentrations among chronic cholecystitis, xanthogranulomatous cholecystitis, and gallbladder cancer patients (GBC) [225]. Cholic acid can be used to differentiate between different types of gallbladder diseases [226]. Extracts of gallstones with GBC present higher concentrations of calcium and magnesium compared to non-cancerous gallbladder inflammation [227].

A metabolome analysis could also be employed in more sophisticated studies. An example was provided by Poland and co-workers to better understand the profile changes in the gut metabolome after a novel surgical procedure in mice, termed distal gallbladder bile diversion to the ileum, that emulates the altered bile flow after Roux-en-Y gastric bypass without other manipulations of the gastrointestinal anatomy. Fecal samples from mice that had undergone bile diversion surgery were analyzed by ultraperformance liquid chromatography–ion mobility–mass spectrometry (UPLC–IM–MS). This method identified a dysregulation of bile acids, short-chain fatty acids, and cholesterol derivatives contributing to the differential metabolism detected [228].

## 7. Conclusions

Food allergies are currently common, in particular in advanced countries, where they are rapidly increasing, already placing a heavy burden on hospitals and other clinical settings. The contribution of the gut microbiota is finally being acknowledged, if not yet fully understood, as the metabolites they produce exert multiple effects on the host’s immune system, controlling the production of immunoglobulins, type A and, specially, type E, with the latter inducing hypersensitivity reactions. Some bacteria can even directly affect the systems (involving occludin and claudins) that control the gut barrier function, hence generating leaks and allowing the passage of either bacteria or their products, which can activate detrimental processes in distal tissues or organs. This can also result in pathogenic microorganisms colonizing parts of the human body, such as in the gallbladder, causing inflammation and resulting in either food allergies or serious infections that require surgery, or can even be lethal. Recent studies indicate that many of the current food allergies can be ameliorated, or even cured, by treatment with either beneficial polyamines or the bacteria that generate them, restoring healthy levels of these compounds, not only in the GI system, but mainly in the blood stream. Although a wealth of knowledge has already been amassed on the topics encompassed in the present review, much more information is still required for humanity to reap the putative health benefits provided by bacteria and their metabolites. One such an example is the tantalizing realization that polyamines, and the bacteria that produce them, could play a major role, not only in preserving brain functions such as memory in elderly people, but also in extending the healthy human lifespan.

## Figures and Tables

**Figure 1 ijms-23-14333-f001:**
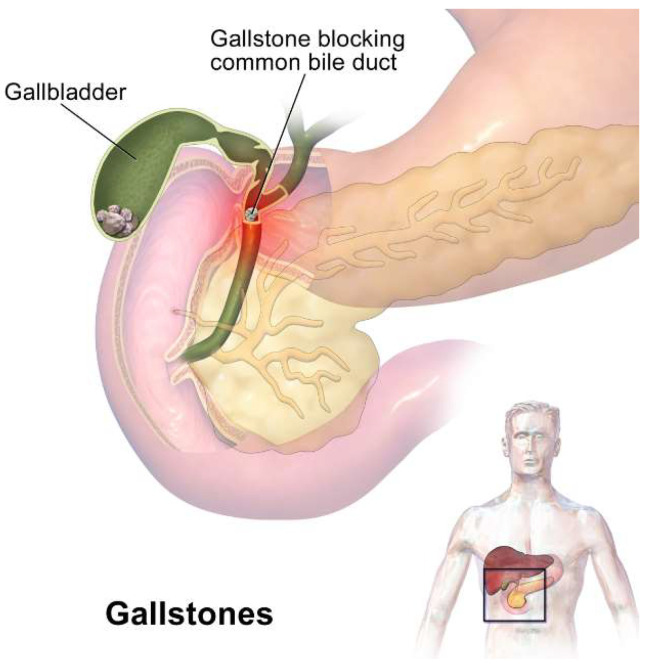
Graphic representation of the human gallbladder and the liver.

**Figure 2 ijms-23-14333-f002:**
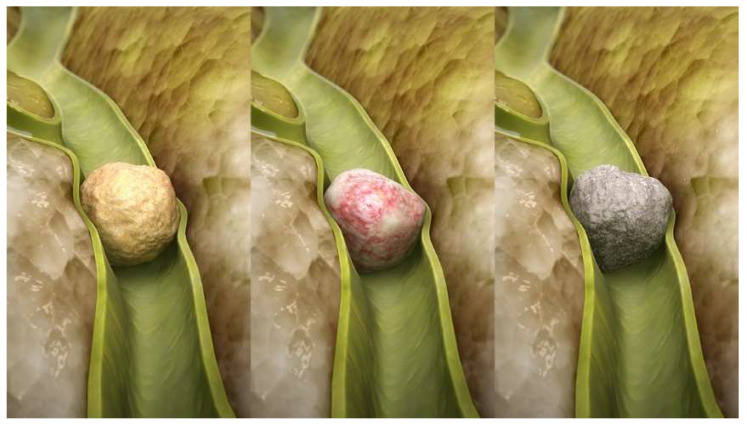
Gallbladder gallstones (calculi). A visual design illustrating the three types of gallstones that can be present in the human gallbladder, classified according to their composition: cholesterol (**left**), pigment (**middle**), and a mixture of both types of gallstones (**right**).

**Figure 3 ijms-23-14333-f003:**
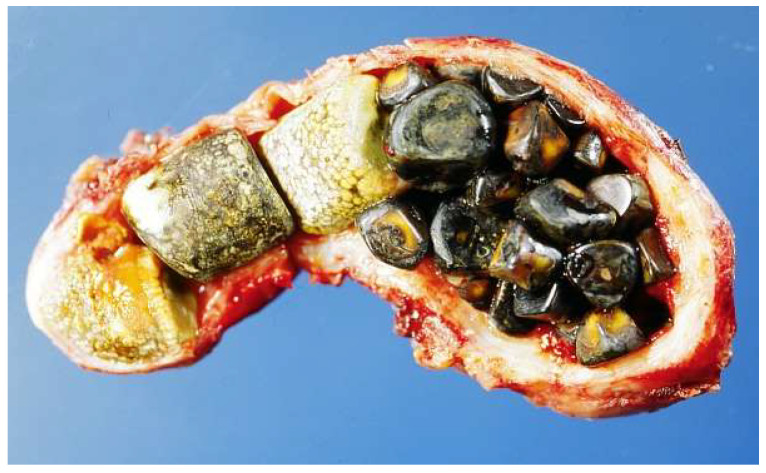
An open gallbladder entirely filled with gallstones (calculi).

**Figure 4 ijms-23-14333-f004:**
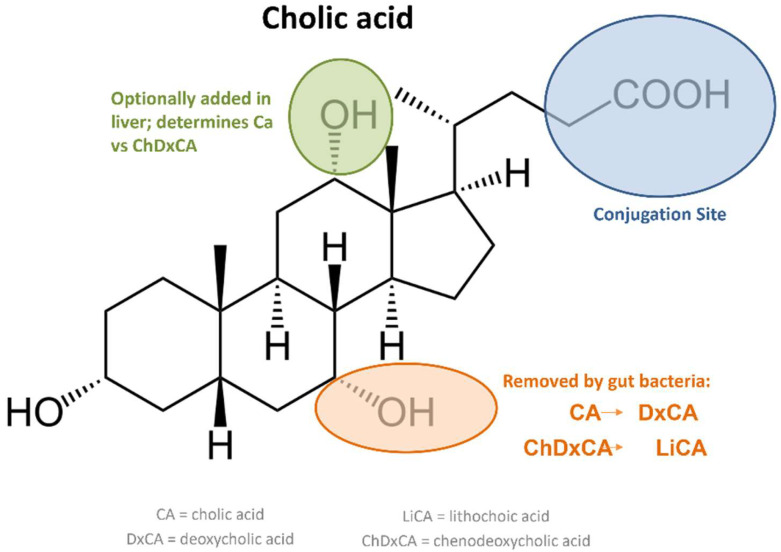
Structure of cholic acid, a primary bile acid, displaying the radicals susceptible to the action of gut bacteria.

**Figure 5 ijms-23-14333-f005:**
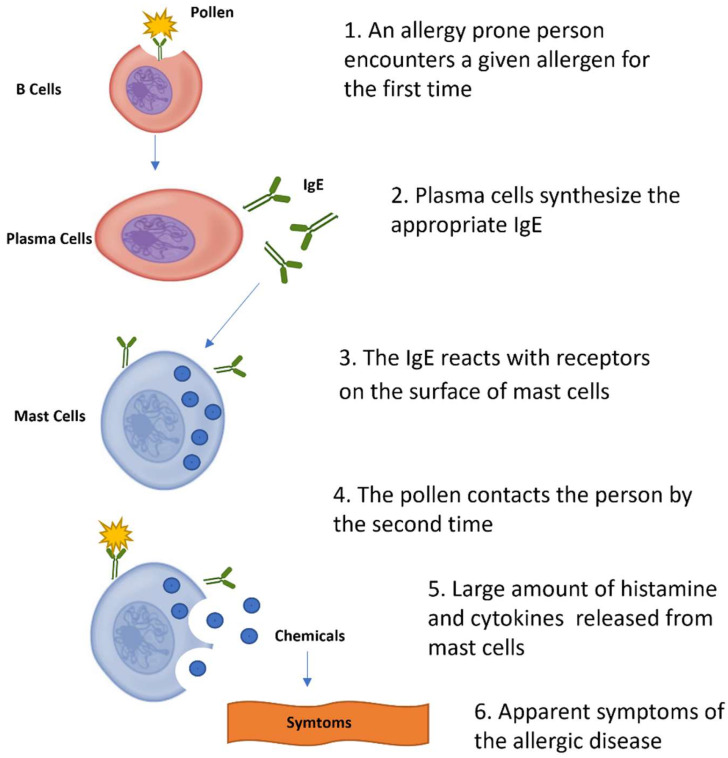
Generation of IgE after ingestion of allergens present in foodstuffs.

**Figure 6 ijms-23-14333-f006:**
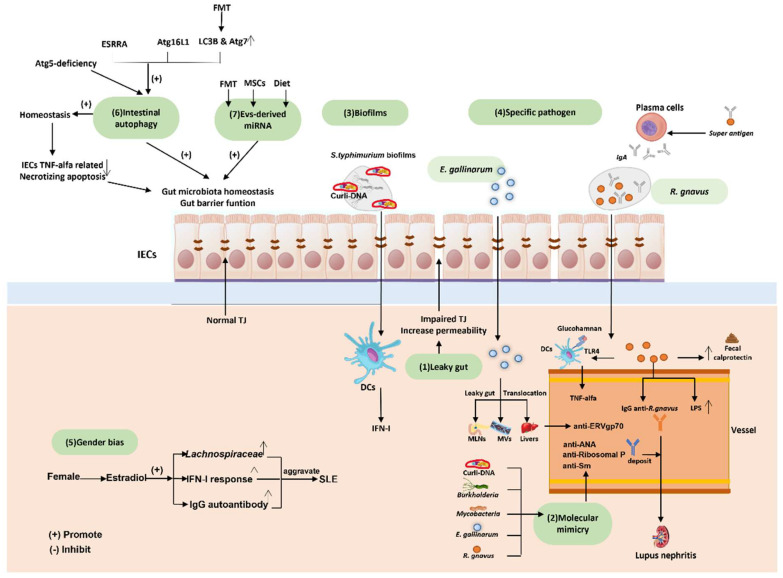
Mechanisms of enteric dysbiosis in SLE. A leaky gut can facilitate the entrance of microorganisms, by either the blood stream or the lymphatic system, and affect distal organs; one such example is the deposition of biofilms on the kidneys. Some bacteria (*Enterococcus gallinarum*) disrupt the intestinal epithelium and translocate to distal organs, such as the liver, which, upon induction of ERV gp70 liver overexpression, can lead to autoimmune syndromes. *Ruminococcus gnavus* displays harmful responses, as it expresses a B-cell superantigen that results in the overproduction of IgA, as well as possessing a capsule that facilitates intestinal colonization. In addition, this bacterium secretes the inflammatory factor TNF-α and produces an antigen that causes the generation of anti-dsDNAgnavus antibodies and aggravates lupus nephritis. ATG, autophagy-related protein; DCs, dendritic cells; *E. gallinarum*, *Enterococcus gallinarum*; ESRRA, estrogen-related receptor alpha; FMT, fecal microbiota transplantation; IECs, intestinal epithelial cells; IFN-I, type I interferon; LC3B, microtubule-associated protein 1 light chain 3B; LPSs, lipopolysaccharides; MLNs, mesenteric lymph nodes; MVs, mesenteric veins; MSCs, mesenchymal stem cells; R. gnavus, Ruminococcus gnavus; TJ, tight junction; TLR4, toll-like receptor 4; TNF-α, tumor necrosis factor-α. This graphic representation is based on the publication by Pan and colleagues in [104] that explains the signs produced by systemic lupus erythematosus.

**Table 1 ijms-23-14333-t001:** Differences in microbial relative abundance (% of sequences) in bile at genus level between cholelithiasis patients and the control group. Only genera with a mean relative abundance higher than 0.5% are presented. Only genera that were detected in more than half of the samples in each group were considered for the analysis. Directly taken from Molinero and colleagues in 2019 [24] with permission from the editorial office and authors.

Genera	Cholelithiasis Group ^a^	Control Group	*p* Value ^b^
*Acidibacter*	0.13 ± 0.33	3.15 ± 4.4	0.005
*Actinobacillus*	3.34 ± 12.03	0.00 ± 0.00	0.225
*Alistipes*	3.85 ± 2.16	1.72 ± 2.31	0.031
*Alloprevotella*	0.53 ± 0.43	0.23 ± 0.35	0.346
*Bacteroides*	10.21 ± 6.94	2.74 ± 3.77	0.001
*Barnesiella*	1.45 ± 0.90	0.73 ± 1.05	0.084
*Bifidobacterium*	3.01 ± 5.02	7.60 ± 17.98	0.599
*Blautia*	1.15 ± 1.22	0.79 ± 1.63	0.56
*Bradyrhizobium*	0.18 ± 0.44	6.90 ± 8.56	0.004
*Brevundimonas*	0.15 ± 0.20	2.80 ± 3.31	0.003
*Christensenellaceae* R.7 group	0.56 ± 0.44	0.08 ± 0.20	0.04
*Coprococcus* 3	1.19 ± 0.69	0.35 ± 0.67	0.009
*Dialister*	1.49 ± 1.40	0.12 ± 0.25	0.002
*Escherichia-Shigella*	4.90 ± 10.52	0.30 ± 0.49	0.001
*Eubacterium comprostanoligenes* group	2.22 ± 2.53	0.59 ± 0.81	0.01
*Faecalibacterium*	2.22 ± 1.31	1.53 ± 3.66	0.661
*Haemophilus*	7.09 ± 25.28	0.02 ± 0.04	0.011
*Haliangium*	0.01 ± 0.04	0.54 ± 0.88	0.006
*Helicobacter*	10.84 ± 8.46	6.29 ± 9.32	0.232
*Lachonospira*	2.75 ± 5.32	0.54 ± 1.32	0.055
*Lachnospiraceae* NK4A136 group	1.41 ± 0.92	0.63 ± 0.88	0.039
*Lactococcus*	0.53 ± 1.10	13.14 ± 24.37	0.007
*Methylobacterium*	0.04 ± 0.04	1.80 ± 3.76	0.003
*Parabacteroides*	0.68 ± 0.43	0.14 ± 0.28	0.067
*Prevotella*	1.16 ± 1.17	1.17 ± 0.57	0.011
*Prevotellaceae NK3B31 group*	0.59 ± 0.56	0.06 ± 0.18	0.04
*Propionibacterium*	0.58 ± 0.40	10.77 ± 18.48	0.01
*Pseudobytyrividrio*	0.59 ± 0.36	1.11 ± 3.20	0.977
*Ruminococcaceae* UCG-002	0.82 ± 0.60	0.17 ± 0.33	0.023
*Ruminococcaceae* UCG-004	0.56 ± 0.34	0.23 ± 0.35	0.229
*Ruminococcus*	0.51 ± 0.48	015 ± 0.42	0.182
*Sediminibacterium*	0.55 ± 0.94	4.65 ± 4.75	0.008
*Sphingomonas*	0.03 ± 0.05	2.74 ± 4.68	0.001
*Streptococcus*	6.67 ± 22.33	0.89 ± 0.81	0.957
*Subdoligranulum*	1.65 ± 1.24	0.47 ± 0.71	0.013
*U.m. of Bacteroidales* S24-7 group family	1.48 ± 1.05	0.78 ± 1.11	0.111
*U.m. of Caulobacteraceae* family	1.14 ± 0.11	2.52 ± 3.14	0.009
*U.m. of Lachnospiraceae* family	6.05 ± 3.05	3.46 ± 4.33	0.108
*U.m. of Ruminococcaceae* family	1.05 ± 0.58	1.09 ± 2.17	0.986

^a^ Mean relative abundance standard deviation. ^b^ Statistical significance was considered with a *p* value below 0.05, adjusted for multiple-hypothesis testing using a false-discovery rate (FDR) correction of 0.25.

**Table 2 ijms-23-14333-t002:** Some bile resistance mechanisms in Gram-negative enteric pathogens. The table was constructed using data from Sistrunk and colleagues in 2016 [53].

Pathogen	Function of Outer Membrane Proteins (Reference[s])	Mechanism of Induction of Stress Response Genes	Efflux Pump (s)
*Escherichia coli*	Repression of *ompF* [54]	*sulA* to correct DNA damage [55]	AcrAB [56]
*Vibrio*	Regulated expression of *ompU* and *ompT* [57]	RpoS in *V. vulnificus* [58]	AcrAB, BreR repressor, vexAB and vexCD, VprAB, Vme pumps [59,60,61,62,63]
*Salmonella*	Repression of OmpF and OmpC, utilization of TolA and TolC [64,65]	SoxRS, OxyR [66,67]	AcrAB [68,69], AcrEF, MdtABC, MdsCBA, EmrAB, MdtK, and MacAB
*Campylobacter*			CmeABC and CmeDEF [70]

## Data Availability

Not applicable.

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
