# Peer review of "The Role of the Gallbladder, the Intestinal Barrier and the Gut Microbiota in the Development of Food Allergies and Other Disorders"

_ijms, 2022, doi:10.3390/ijms232214333_

Round 1

Reviewer 1 Report

A well-written article about the intestinal microflora, food allergies and other autoimmune conditions. The authors described various health conditions and severe diseases that are influenced by the intestinal flora or caused by obstruction and inflammation of the gallbladder, and putative treatments for these diseases.

A detailed study of the intestinal microflora and the metabolites produced by the bacteria shows how they affect the host's immune system. Some bacteria can even directly affect the systems that control the intestinal barrier. Some of the studies described indicate that many of the food allergies present can be alleviated and even cured by treatment with beneficial polyamines or bacteria.

The literature review contains a lot of items, most of them are articles from the last 10 years.

Author Response

Thank you very much for your revision. We appreciate it.

Kind regards

Ana

Reviewer 2 Report

This is a review on the role of the gut microbiota in the development of immune disorders.

Authors have tried to provide a connection between the gallbladder with allergy and autoimmune disorders, but have offered only an extensive descriprion of the gallbladder diseases (pages 4 and 5) without any obvious reason.

The IgE-secretion has been presented as the cornerstone of immune-mediated effects of gut microbiota in page 14, failing to underline other mechanisms of native immunity that are caused by an interaction with the gut microbiota.

An approach of the leaky gut syndrome has been made, with references on tight junctions, while the impact of other disorders of cell junctions, namely adherence junctions and desmosomes that are also affecting the microbiota-immunity interaction, are not adequately mentioned.

In order to provide this review, the authors are offerering a lot of information on anatomy or physiology that are -in my opinion- unecessary and make it exteremely difficult for the reader to focus on the aim of this review.

Some inaccuracies are also found, like the mention of coeliac disease  as food allergy (line 73) and the introduction of breast milk (line 140) in the paragraph on lymph flow, causing a confusion on the relation of these two separate topics.

I am afraid I do not think that this review is not appropriate for pubblication and I would advice the authors to make a strict selection on the parts regarding microbiota that affects the related disorders of the  immune system.

Author Response

Thank you very much for your opinion. In order to improve the manuscript, your following suggestions have been taken into consideration and answered to the best of our knowledge with detailed proof-reading.

Revisor 2 comments and suggestions:

This is a review on the role of the gut microbiota in the development of immune disorders.

Authors have tried to provide a connection between the gallbladder with allergy and autoimmune disorders, but have offered only an extensive descriprion of the gallbladder diseases (pages 4 and 5) without any obvious reason.

Answer: In reference to the first point raised by this referee, the fact that the gallbladder can be a reservoir of microorganisms (including pathogenic ones), particularly in cases of duct occlusion by stones or mal-functioning of the Oddi´s sphincter, could be treated as a good reason to be included in the actual review, and undoubtedly related with actual gut´s dysbiosis. 

The IgE-secretion has been presented as the cornerstone of immune-mediated effects of gut microbiota in page 14, failing to underline other mechanisms of native immunity that are caused by an interaction with the gut microbiota.

Answer: As for the second paragraph on the role of IgEs, we are aware that other mechanisms of the native immunity may be involved, which was included in the manuscript under the "umbrella" of  ‘non-IgE-mediated’ allergic reactions, often a term used to avoid a time-consuming description. We believe that the actual paragraph does reflect this idea.

An approach of the leaky gut syndrome has been made, with references on tight junctions, while the impact of other disorders of cell junctions, namely adherence junctions and desmosomes that are also affecting the microbiota-immunity interaction, are not adequately mentioned.

Answer: As for the third point of concern, we believe that we have resolved by leading the future reader to the following reference: " Gierynska, M.; Szulc-D ?abrowska, L.; Struzik, J.; Mielcarska, M.B.; GregorczykZboroch, K.P. Integrity of the Intestinal Barrier: The Involvement of Epithelial Cells and Microbiota—A Mutual Relationship. Animals 2022, 12, 145".  Where many aspects concerning the integrity of the gut´s epithelium are dealt with, in an appropriate manner.

In order to provide this review, the authors are offerering a lot of information on anatomy or physiology that are -in my opinion- unecessary and make it exteremely difficult for the reader to focus on the aim of this review.

Some inaccuracies are also found, like the mention of coeliac disease  as food allergy (line 73) and the introduction of breast milk (line 140) in the paragraph on lymph flow, causing a confusion on the relation of these two separate topics.

Answer: In reference to the lack of accuracy concerning the origin of the celiac disease in the first Ms, has been corrected now with the phrase " is a long-term autoimmune disorder".

As for the breast milk concern, this has been resolved by suppressing the paragraph, and by doing so we think the inaccuracy no longer exists

I am afraid I do not think that this review is not appropriate for pubblication and I would advice the authors to make a strict selection on the parts regarding microbiota that affects the related disorders of the  immune system.

Reviewer 3 Report

In the manuscript “The role of the gallbladder, the intestinal barrier and the gut microbiota in the development of food allergies and other disorders”, the authors investigated the role of gut microbiota and gallbladder in microbial infections and autoimmune disorders such as lupus erythematous and allergies. In particular, they focused on food allergies.

In my opinion, the manuscript is well-written and deals with an interesting thematic. I encourage the publication after addressing the following points:

1)    In paragraph 3, I suggest the authors to write briefly about the contribution of bacteria- after gallbladder invasion- in producing pro-inflammatory factors, responsible for inflammatory events. 

2)    I also encourage authors to introduce some lines about the role of inflammation and dysbiosis in tumor progression.

3)    I suggest improving the quality of Figure 6.

4)    In paragraph 6, authors write about metagenomics, proteomics and transcriptomics applied to the gallbladder study. What about metabolomics?

Author Response

Answer: Thank you very much for your opinion. In order to improve the manuscript your following suggestions have been taken into consideration and answered to the best of our knowledge with detailed proof-reading.

  • In paragraph 3, I suggest the authors to write briefly about the contribution of bacteria- after gallbladder invasion- in producing pro-inflammatory factors, responsible for inflammatory events. 

Answer: According to the referee's suggestion we wrote some lines about the contribution of bacteria- after gallbladder invasion- in producing pro-inflammatory factors, responsible for inflammatory events.

"Pathogen species, such as enterotoxigenic Bacteroides fragilis indeed does remodel the colonic microbiota thus promoting colorectal cancer, possibly via IL-17 by TH17 cell-mediated inflammation. The process however could also started in a microbiologically independent manner and based, but in the end suppressed by the existing beneficial commensal microbiota [Sheflin et al.2014]."

  • I also encourage authors to introduce some lines about the role of inflammation and dysbiosis in tumor progression.

Answer: Following the referee's recommendation, we have introduced some lines about the role of inflammation and dysbiosis in tumor progression:

"Salmonella infection has an important role in gallbladder cancer [Sheflin et al.2014] as gallbladder is a reservoir for some species of this genus.  Salmonella gallbladder infection increases in secondary bile acid (SBA) leading to gallbladder carcinogenesis [Shukla et al.]. Gallbladder cancer is associated with the typhoid carrier state and positivity for the Vi antigen (a capsular antigen associated with Salmonella). This relatedness shows the relevance of the microbiome in the pathogenesis of multiple cancer types (Sharma et al 2007)".

  1. Sheflin, A. M., Whitney, A. K., & Weir, T. L. (2014). Cancer-promoting effects of microbial dysbiosis. Current oncology reports16(10), 1-9.
  2. Sharma V, Chauhan VS, Nath G, Kumar A, Shukla VK. Role of bile bacteria in gallbladder carcinoma. Hepatogastroenterology. 2007;54(78):1622.
  3. Shukla V, Tiwari S, Roy S. Biliary bile acids in cholelithiasis and carcinoma of the gall bladder. Eur J Cancer Prev. 1993;2(2):155–60.
  • I suggest improving the quality of Figure 6.

Answer: The quality of Figure 6 has been improved. However, the png file quality is better than the quality of the figure inserted in the manuscript document.

  • In paragraph 6, authors write about metagenomics, proteomics and transcriptomics applied to the gallbladder study. What about metabolomics?

Answer:  We have followed the referee's recommendations and we have added recent reports about metabolomics applied to the gallbladder study to improve the manuscript.

"In addition, many studies related to gallbladder have been provided by metabolomics. Common methods used in metabolomics are mass spectrometry and nuclear magnetic resonance (NMR) spectroscopy [221]. NMR technology has been employed to analyze the lipid profiles of gallbladder tissue extracts, with the outcome that alterations in the lipid metabolism may be detected upon comparison between benign and malignant gallbladder diseases [222]. Additional researchs based on NMR analysis did report on variations in concentrations of low-density lipids (LDL), very-low-density lipids (VLDL), branched chain amino acids (BCAA), alanine, glutamine, creatinine, and tyrosine, 1,2-propanediol, pyruvate, glutamate, and formate in gallbladder cancer as compared to healthy individuals [223]. Moreover, gallbladder stones have been analyzed using solid-state NMR from patients from different geographical regions showing different levels of cholesterol, bilirubin, and calcium carbonate correlations [224].

NMR technology has provided as well useful information about differences in metabolite concentrations among chronic cholecystitis, xanthogranulomatous cholecystitis, and Gallbladder cancer patients (GBC) [225]. Cholic acid can be used to differentiate between different types of gallbladder diseases [226]. Extracts of gallstones with GBC present higher concentrations of calcium and magnesium compared to non-cancerous gallbladder inflammation [227].

Metabolome analysis could be also employed in more sophisticated studies. An example was provided by Poland and coworkers to better understand the profile changes in the Gut metabolome after a novel surgical procedure in mice, termed distal gallbladder bile diversion to the ileum, that emulates the altered bile flow after Roux-en-Y gastric bypass without other manipulations of gastrointestinal anatomy.  Fecal samples from mice that have undergone bile diversion surgery were analyzed by ultraperformance liquid chromatography-ion mobility-mass spectrometry (UPLC-IM-MS). This method has identified a dysregulation of bile acids, short-chain fatty acids, and cholesterol derivatives contributing to the differential metabolism detected [228]".

Round 2

Reviewer 2 Report

Authors have chosen a very interesting topic and provided a slightly improved revised version. However, I feel like the final draft cannot be accepted in this journal.

Author Response

Thank you very much for your opinion. We will keep working.